# The Journey of Copper-Impregnated Dressings in Wound Healing: From a Medical Hypothesis to Clinical Practice

**DOI:** 10.3390/biomedicines13030562

**Published:** 2025-02-24

**Authors:** Gadi Borkow, Eyal Melamed

**Affiliations:** 1MedCu Technologies Ltd., Herzliya 4672200, Israel; 2The Skin Research Institute, The Dead-Sea & Arava Science Center, Masada 8691000, Israel; 3Foot and Ankle Service, Department of Orthopedics, Bnai Zion Medical Center, Haifa 3109601, Israel

**Keywords:** copper oxide, wound dressings, wound healing, angiogenesis, extracellular matrix, chronic wounds

## Abstract

**Background/Objectives.** Chronic wounds pose a substantial global healthcare burden exacerbated by aging populations and the increasing prevalence of conditions such as diabetes, peripheral vascular disease, and venous insufficiency. Impaired physiological repair mechanisms, including angiogenesis, collagen synthesis, and re-epithelialization, hinder the healing process in chronic wounds. Many of these physiological processes are dependent on their interaction with copper. We hypothesized that the targeted delivery of copper ions to the wound bed would enhance healing. **Methods.** Wound dressings impregnated with copper oxide microparticles were designed to ensure the controlled release of copper ions. The efficacy of these dressings was evaluated using non-infected wound models, including diabetic mouse models compared against control and silver dressings. Outcome measures included wound closure rates, epidermal skin quality assessed by histopathological examination, and gene expression profiling. Clinical applications were assessed through diverse case studies and controlled trials involving chronic wound management. **Results.** Copper dressings significantly accelerated wound closure and enhanced angiogenesis compared to control and silver dressings. Histopathological analyses revealed faster granulation tissue formation, epidermal regeneration, and neovascularization. Gene expression studies showed upregulation of critical angiogenic factors such as VEGF and HIF-1α. Investigations and clinical observations corroborated improved healing across various chronic wound types, including non-infected wounds. **Conclusions.** Copper is essential for wound healing, and copper-impregnated dressings provide a promising solution for chronic wound management. By enhancing angiogenesis and tissue regeneration, these dressings go beyond antimicrobial action, offering a cost-effective and innovative alternative to conventional therapies. Copper dressings represent a transformative advancement in addressing the challenges of chronic wound care.

## 1. Chronic Wounds

Chronic wounds, defined as wounds that fail to proceed through an orderly and timely reparative process, affect a significant portion of the world population. As the life expectancy of the worldwide population is increasing, the prevalence of chronic wounds, such as venous ulcers, pressure sores, and diabetic ulcers, is also dramatically increasing, particularly in developed countries. The global prevalence of venous ulcers is estimated to be around 1% of the general population, increasing to approximately 3% in individuals over 65 years of age [1]. A 2019 study estimated the global diabetes prevalence at 9.3% (463 million people), which is projected to rise to 10.2% (578 million) by 2030 and 10.9% (700 million) by 2045 [2]. A meta-analysis by Li et al. [3] reported a pooled global prevalence rate of 12.8% for pressure ulcers, indicating that approximately one in eight individuals in healthcare settings may be affected. A meta-analysis of studies analyzing chronic wounds in the general population estimated a pooled prevalence of 2.21 per 1000 people in the population for chronic wounds of mixed etiologies, and 1.51 per 1000 for chronic leg ulcers [4]. In the United States, chronic wounds impact the quality of life of nearly 2.5% of the total population, with a higher prevalence observed among the elderly [5]. In Germany, in 2012, approximately 1.04% of insured patients had a wound diagnosis, with 0.70% having leg ulcers and 0.27% diabetic ulcers [6]. Extrapolated to the national level, this equates to about 786,407 individuals with chronic wounds. Taken together, it is clear that chronic wounds are an alarming increasing global problem, causing enormous cost burdens to both patients and healthcare providers [7].

The healing of wounds follows a precisely regulated and systematic process consisting of three overlapping phases after hemostasis: inflammation (lasting 4 to 6 days), proliferation (spanning 4 to 24 days), and remodeling (which can extend from 21 days up to 2 years). These biological events involve various cellular and molecular mechanisms, including cell migration, differentiation, and proliferation. When a wound occurs, changes in the microenvironment, such as decreased oxygen levels, trigger the release of specific factors from fibroblasts, epidermal cells, macrophages, and vascular endothelial cells, promoting the formation of new blood vessels (angiogenesis). Among these factors, vascular endothelial growth factor (VEGF) plays a dominant role in driving angiogenesis, while Transforming Growth Factor-beta (TGF-β), particularly TGF-β1, is crucial for regulating the production of elastin and collagen. Fibroblasts synthesize and secrete these structural proteins, which then undergo cross-linking to form the extracellular matrix (ECM) of the dermis. This process is essential for restoring the structural integrity and function of the damaged tissue.

Chronic wounds, also referred to as “hard-to-heal wounds”, become stalled in one or more stages of the healing process, with some taking years to close completely or failing to heal altogether. One of the major processes that do not occur in chronic wounds is the production of new blood capillaries. Chronic wounds are characterized by extensive loss of the integument, necrosis, or signs of circulation impairment. These wounds are associated with high morbidity and mortality due to tissue inflammation, ischemia, and infection [8,9,10].

Wound infections delay wound healing [11] by producing inflammatory mediators, metabolic wastes, and toxins; causing tissue hypoxia and hemorrhagic and fragile granulation tissue; reducing fibroblast number and total collagen production; interfering with re-epithelialization [12,13]; and reducing the available nutrients and oxygen needed by the host cells and causing neutrophils to be in an activated state producing cytolytic enzymes and free oxygen radicals [14]. Bacteria are usually protected by biofilm in chronic wounds [11]. Thus, reducing the microbial contamination of wounds increases the capacity of the wound to heal.

## 2. The Role of Copper in Wound Healing

Copper is a vital trace element that plays a role in numerous physiological functions across all body tissues, including the skin and integumentary system [15,16,17,18]. Several crucial wound healing mechanisms rely on copper for proper function [19,20]. These include the regulation of key growth factors such as VEGF and angiogenin, which drive angiogenesis during the proliferation phase [21,22,23,24,25]; the production of collagen (types I, II, and V) and elastin fibers by dermal fibroblasts, which are essential for tissue formation in the proliferation and remodeling phases [26,27,28]; and the activity of lysyl oxidase (LOX), which facilitates cross-linking between collagen and elastin, ensuring the stability of the ECM [29,30,31]. Additionally, copper influences integrin modulation via differentiated keratinocytes in the remodeling phase and is involved in the function of matrix metalloproteinases (MMPs) and serine proteases, such as human neutrophil elastase (HNE), which contribute to the wound healing process [32,33,34,35,36]. As a cofactor of superoxide dismutase [23], an antioxidant enzyme present in the skin, copper helps mitigate oxidative stress by preventing membrane damage and lipid peroxidation. Copper is also a cofactor of tyrosinase [37], the enzyme responsible for melanin production, which plays a role in skin and hair pigmentation, particularly during the maturation phase of wound healing. The essentiality of copper in the natural physiological wound healing processes is now very well established [20].

Other natural trace elements play an important role in wound healing. For example, cobalt is an essential component of vitamin B12 (cobalamin), which is crucial for red blood cell formation, DNA synthesis, and cellular metabolism—important factors in tissue repair. Additionally, cobalt has been shown to stimulate angiogenesis. Recent research has explored the use of cobalt-releasing bioactive glass compositions as scaffold materials to locally deliver pro-angiogenic cobalt ions at a controlled rate, aiming to promote wound healing [38,39].

## 3. Copper as a Wide Spectrum Antimicrobial Tool

Copper is also an essential mineral required for the normal functioning of microorganisms [40]. However, microorganisms must carefully regulate intracellular copper levels. Under anaerobic conditions, copper exists predominantly in its highly reactive cuprous form (Cu^+^), which readily interacts with microbial proteins, disrupting their structure by forming thiolate bonds with iron–sulfur clusters [41]. When microorganisms are exposed to excessive concentrations of copper, they are unable to manage the surplus, leading to their death [42,43].

The potent biocidal activity of copper is attributed to several mechanisms. It is likely that the first site damaged by copper is the microorganism’s envelope [42]. Significant copper-induced disruption of membrane integrity ultimately results in a loss of cell viability. However, even minor changes in the physical properties of biological membranes can profoundly affect essential membrane-dependent functions, such as transport protein activity and ion permeability [44]. Once inside the cell, copper can displace essential metals from their native binding sites within proteins or interact directly with the proteins. The redox cycling between Cu^+^ and Cu^2+^ generates hydroxyl radicals, which preferentially target amino acids such as histidine and proline. This process can cause significant protein alterations, including structural modifications and, in some cases, protein cleavage [45,46]. These disruptions often result in conformational changes to the protein structure or active site, ultimately inhibiting or neutralizing the proteins’ biological functions. Copper ions may also interact with nucleic acids, especially in viruses, causing damage to the genetic material [42]. Copper’s multisite, non-specific mechanism of action, which simultaneously targets and damages multiple critical components of microorganisms, makes the development of resistance to copper exceedingly difficult [42]. In stark contrast to the widespread resistance microorganisms have developed to antibiotics within just 50 years of their use, resistance to copper remains exceptionally rare, despite copper’s presence on Earth for millions of years.

## 4. Do Chronic Wounds Not Heal Due to Insufficient Local Copper Levels?

In 2008, we hypothesized that chronic wounds may not heal due to the lack of systemic copper that reaches the wound [19]. We based our hypothesis on the key role that copper plays in the main wound healing processes, i.e., induction of angiogenesis, secretion of extracellular matrix skin proteins, and epithelialization. We hypothesized that in individuals with diabetic ulcers, pressure sores, peripheral vascular wounds, or other injuries with potentially compromised circulation to the wound site, a contributing factor to impaired healing may be insufficient local copper levels. We further theorized that constant in situ exposure of copper ions to the wound site would enhance wound repair by inducing angiogenesis and upregulating specific elements that are critical for the healing process and are stagnated in chronic wounds [19]. We envisioned that incorporating copper into wound dressings could not only minimize the risk of wound and dressing contamination, similar to silver, but also actively promote faster wound healing. This effect would be achieved through the controlled release of copper from the dressings directly into the wound site, stimulating angiogenesis and facilitating skin regeneration.

To validate our hypothesis, we developed wound dressings impregnated with cuprous oxide microparticles [47]. On the one hand, we endowed the dressings with potent biocidal properties [47,48], and on the other hand, these dressings served as a sustained-release reservoir for copper ions and are capable of continuously delivering copper ions to the wound bed, without causing any adverse reactions [47]. We then evaluated the dressings in aseptic wound models using genetically engineered diabetic mice. Wounds were treated with either control dressings (lacking copper), commercially available silver dressings, or copper oxide-impregnated dressings. The mice treated with copper oxide-impregnated dressings exhibited a statistically significant acceleration in non-infected wound closure compared to those treated with silver or control dressings [49]. Histopathological analysis of skin specimens from the mice treated with copper dressings revealed normal histology with no evidence of atypia. Furthermore, the copper-treated wounds displayed earlier epidermal regeneration, enhanced granulation tissue formation, increased angiogenesis, and the emergence of new hair follicles and sebaceous glands.

Immunohistochemistry staining provided clear evidence of significantly enhanced angiogenesis in the copper-treated mice. Further investigation of mRNA expression levels in wound sites, analyzing 84 genes alongside immunohistochemistry staining, demonstrated the upregulation of multiple angiogenic factors, including placental growth factor (PGLF), vascular endothelial growth factor (VEGF), and hypoxia-inducible factor-1 alpha (HIF-1α). A proposed molecular mechanism suggests that the redox cycling between Cu^+^ and Cu^2+^ creates a hypoxic environment, triggering the increased expression of HIF-1α in the dermis. This upregulation of HIF-1α then initiates a cascade of biological processes, promoting endothelial cell proliferation and migration, angiogenesis, immune cell infiltration, fibroblast activation, heightened metabolic activity, enhanced extracellular matrix protein secretion, and accelerated epithelialization [49].

Hypoxia-inducible factor-1 (HIF-1) deficiency and subsequent failure to respond to hypoxic stimuli leads to chronic hypoxia, which has been shown to contribute to the formation of non-healing ulcers [50]. HIF-1α, a subunit of the transcription factor hypoxia-inducible factor-1 (HIF-1), is now accepted as an important target for the enhancement of wound healing [50,51,52]. Copper is required for HIF-1 activation through the regulation of HIF-1α binding to the hypoxia-responsive element (HRE) and the formation of the HIF-1 transcriptional complex [53]. It has also been demonstrated that copper regulates HIF-1 transcription activity by affecting the selectivity of HIF-1α binding to the promoters of affected genes [54]. HIF-1α is not responsive in diabetics [55,56], as hyperglycemia impairs the HIF-1a transactivation, resulting in decreased hypoxia-induced VEGF expression [57]. HIF-1 abnormal expression also probably occurs in other chronic wounds, as it occurs in other diseases (e.g., rheumatoid arthritis [58]).

Das et al. [59] confirmed our findings by investigating the role of the copper-dependent transcription factor and copper chaperone Antioxidant-1 (Atox1), which maintain intracellular copper homeostasis and play a role in wound healing. They demonstrated that topical treatment with copper of cutaneous wounds inflicted in wild-type mice accelerated wound closure. Furthermore, they found that treatment of the wounds with a copper chelator delayed the healing process. They also found that wounding markedly increased the expression of Atox1, but not of other copper chaperons. Taken together, their studies demonstrated how exogenous and endogenous copper accelerates wound healing via Atox1. Coger et al. found an immediate increase in the concentration of copper in a full-thickness open-wound rat model following wounding of ~40% within 2 days [60]. Lansdown et al. found a marginal increase in the copper concentration in the 2 days following wounding in a similar wound rat study [61]. These studies indicate that part of the normal physiological response to wounding is to deliver more copper to the wound. Yi et al. found that SLC31A1, a copper transporter, was significantly downregulated in refractory diabetic foot ulcers (DFUs) as compared to healthy controls and acute traumatic wounds [62]. This study supports our hypothesis that less copper is delivered to chronic wounds, which impairs the key wound healing processes. Yadav et al. found that the mean serum copper levels in diabetes patients with foot ulcers were significantly decreased as compared to diabetic patients without ulcers [63]. While not direct proof, this finding supports the hypothesis that reduced copper serum concentration in diabetic patients may impede wound healing, resulting in chronic foot ulcers.

## 5. Further Laboratory and Animal Studies Supporting the Key Role That Copper Plays in Wound Healing

In recent years, an increasing number of studies have explored copper-containing biomaterials, particularly those incorporating copper nanoparticles (less than 100 nm in diameter) into polymeric scaffolds and hydrogels, for wound healing applications [64,65,66,67,68,69,70]. These studies consistently demonstrate enhanced healing of non-infected wounds [71,72,73,74,75,76], including those in diabetic [77,78,79,80,81,82,83] and burn-related [84,85,86,87,88,89,90] animal models when compared to control materials. Additionally, copper-containing biomaterials, mostly wound dressings, have shown significant potential in promoting the healing of infected wounds [68,87,91,92,93,94,95,96,97,98], including those infected with MRSA [99] and *Escherichia coli* biofilm [100]. Notably, they have also been effective in diabetic wound models infected with *Staphylococcus aureus* [83,101,102] and MRSA [103]. Recently, due to their large surface area, fast degradation, and transformation into hydroxyapatite (HA), as well as their ease of handling and adaptability in shape, bioactive glass microfibers doped with copper or cobalt have gained increasing attention to serve as a substrate for their potential in promoting wound healing [38,39,69,70]. One of the notorious effects noticed in some of these studies was the significant increase in VEGF and the number of blood vessels in the copper-treated wounds as compared to the controls [72,73,75,77,78,84,97,103]. Many of these studies also demonstrated other increased wound healing processes throughout the wound healing phases, such as increased extracellular matrix deposition, increased nerve regeneration, hair follicle regeneration, and fibroblast and endothelial cell migration and epithelialization and reduced inflammation in the groups of animals treated with the copper-containing biomaterials (e.g., [72,80,82,94,97,103]). Copper-eluting fibers have been shown to enhance the approximation and healing of incisional wounds, including increased angiogenesis around the incision site [104].

It is now well accepted by the scientific community that copper plays a key role in the wound healing processes [20] and systematic reviews of the literature clearly support the notion that in situ exposure of wounds by copper-containing biomaterials enhances the wound healing processes (e.g., [67,105,106]). Despite the vast number of studies showing the potential of enhancing wound healing through copper-containing dressings, to the best of our knowledge, the only copper-containing wound dressings currently in clinical use for human wound management are those produced by MedCu Technologies Ltd. (www.medcu.com, Accessed on 1 January 2025). These dressings (MedCu Antimicrobial Wound Dressings with Copper-Oxide; hereafter referred to as copper dressings) containing cuprous oxide microparticles have successfully passed comprehensive biocompatibility and safety evaluations, and have been approved by multiple regulatory bodies, including the FDA and CE, for the treatment of acute and chronic wounds. Commercially launched in 2019, these dressings are now in clinical use in over 25 countries.

## 6. Clinical Applications of Copper Dressings

Numerous studies have demonstrated the ability of copper dressings to enhance healing in hard-to-heal wounds, including both infected and non-infected wounds, across all stages of the healing process [107,108,109,110,111,112,113,114,115,116,117,118,119,120,121,122]. These studies covered a range of chronic wounds, such as diabetic ulcers, venous ulcers, pressure sores, venous harvesting site wounds, non-healing post-amputation wounds, necrotic wounds, and acute conditions like burns and clean orthopedic surgery wounds. Copper dressings were effective even in patients with challenging conditions such as diabetes mellitus, renal failure, inflammatory diseases, and sickle cell disease, as well as in locally compromised tissue due to ischemia, post-irradiation effects, extensive necrosis, and/or contamination. Across these studies, copper dressing application consistently led to significant improvements in wound healing, including in stagnant, non-infected, and non-healing wounds [109,110,112,113,114,115,119,120,121]. An example of such a study was the treatment of diabetic non-infected wounds (wound area range between 1.35 and 23.6 cm^2^) where, during the screening period, the wounds were non-responsive to the standard of care wound dressing management. However, when the copper dressings were applied twice weekly for a month, there was a 53.2% mean wound area reduction (*p* = 0.003) (Figure 1) and a 43.37% increase in mean granulation tissue formation (*p* < 0.001) [110].

The study by Gorel et al. [113] conducted in a rehabilitation center, in 2–30 cm^2^ non-infected wounds treated for 17–41 days with silver wound dressings that failed to reduce the wound size by >50%, is another relevant example. Ten patients were diabetics, ten suffered from hypertension, and six suffered from peripheral vascular disease (PVD). Once they switched the silver dressings to copper dressings, comparing during a period of 25 days, the mean wound area reduction was ~2.4 times higher during the copper treatment phase than during the silver dressing treatment phase (87% versus 37% (*p* < 0.001)). Some of the patients’ wounds responded quite immediately to the exposure to the Copper dressings and their size was reduced quickly (Figure 2).

These studies indicate that the wound healing benefits of copper dressings extend beyond their antimicrobial properties.

Clinical cases have demonstrated the effectiveness of copper dressings in several aspects of wound care, including infection clearance [107,109,111], induction of granulation and epithelialization in necrotic wounds [107,109,111], reduction in post-operative swelling and inflammation, and minimization of scar formation [107]. Notably, these dressings have shown positive outcomes across all stages of wound healing [107,108], supporting a continuum of care from skin rupture to complete closure [108]. The use of copper-containing dressings should be avoided in cases of known allergy to copper or in oncological active wounds.

A recent randomized clinical trial demonstrated that the copper dressings were non-inferior to negative pressure wound therapy (NPWT) in achieving wound size reduction and promoting granulation tissue formation in diabetic ulcers [122]. More specifically, 46 diabetic patients, with non-infected wounds ranging between 1.56 and 44.56 cm^2^, who necessitated NPWT for closure after recovering from wound infections that required extensive debridement surgery and/or partial foot amputation, were randomly allocated to either receiving NPWT or copper dressings. The wound measurement and condition assessed for 13 weeks using a 3D Wound Imaging System App demonstrated no statistically significant differences between both arms in wound size reduction and granulation tissue formation. Moreover, the use of copper dressings was significantly more convenient for both patients and healthcare providers, reducing wound management costs to approximately 15% of those associated with NPWT.

## 7. Case Samples

### 7.1. Clearance of Infection, Induction of Granulation, and Epithelialization [107]

A 57-year-old male with a history of non-insulin-dependent diabetes mellitus (NIDDM) presented with ulcers on both feet, predominantly affecting the right side. The primary cause was attributed to an acute leukocytoclastic vasculitis reaction, with minimal involvement of major arteries. To address this, an angiographic procedure was performed, including percutaneous reopening of the superficial femoral artery. Treatment consisted of high-dose corticosteroids, immunosuppressive therapy (Azathioprine and Imuran), and broad-spectrum antibiotics. Despite these interventions, the condition of the right foot deteriorated, leading to necrosis primarily affecting the medial toes. The infection advanced, compromising the tendons and plantar fascia. Additionally, deep ulcers developed on the medial side of the heel and the lateral part of the foot (Figure 3a,b). The patient underwent surgical debridement, including the amputation of the first and second rays. Intraoperative cultures identified *Pseudomonas aeruginosa* as resistant to quinolones, prompting treatment with Imipenem. However, five days later, the patient required a trans-metatarsal amputation. The wound was only partially closed to avoid a loose flap, but within days, necrosis developed at the flap edges (Figure 3c). Bedside debridement was performed, and cultures from this second surgery revealed that the *Pseudomonas* had developed resistance to carbapenems, leading to the discontinuation of antibiotic therapy. At this point, approximately 30% of the medial heel wound and 80–90% of the lateral anterior wound were necrotic (Figure 3c,d).

A trans-tibial amputation was considered the next step. The patient’s white blood cell count had decreased to 18,000, an improvement from previous levels, and C-reactive protein (CRP) was measured at 3.0 (normal <0.5). However, given the patient’s overall stable condition, it was decided to proceed with local wound care using copper dressings. Copper dressings were applied deep in the plantar fascia portion of the amputation wound, along the wound edges, and over the ulcers (Figure 3d,e). These were changed twice weekly, with Prontosan^®^ irrigation recommended during dressing changes. No additional antibiotic therapy was administered.

Gradual improvement in the foot’s condition was observed. The superficial semi-necrotic ulcer on the heel and lateral foot progressively absorbed necrotic tissue, forming granulation tissue and epithelializing (Figure 3f–i). The primary amputation wound, which had a significant cavity of 6–7 cm, slowly filled with granulation tissue (Figure 3g). Autolysis (self-debridement) of necrotic tissue was also noted (Figure 3i), and new epithelial layers progressively covered the healing wounds (Figure 3g,h). A microbial culture taken three months after stopping antibiotics showed no presence of *Pseudomonas*, though non-pathogenic colonizing bacteria were detected.

After five months of copper dressing treatment, both the medial and lateral wounds had fully closed (Figure 3j,k). The primary amputation site was partially closed, with the remaining wound area covered in healthy pink to red granulation tissue (Figure 3l).

### 7.2. Promotion of Epithelialization [107]

The remarkable ability of copper dressings to enhance epithelialization is demonstrated in the case of a 71-year-old man with NIDDM and diabetic neuropathy. He developed osteomyelitis of the calcaneus, requiring extensive debridement of both the heel and the infected bone. However, the wound failed to heal, and due to the missing bone, the calcaneus collapsed through a weakened area, resulting in a rocker-bottom deformity. To address this, the patient underwent another surgical procedure involving soft tissue debridement, realignment of the foot, and fixation with Steinman pins (Figure 4a,b, one week post-surgery).

During and after surgery, the wound was treated with copper dressings, which were changed weekly. Rapid granulation tissue formation was observed, filling both the depth and walls of the large wound cavity. Simultaneously, healthy skin from the wound edges began migrating inward. A photograph taken three weeks post-surgery (Figure 4c) shows early epithelialization at the superficial cavity wall (indicated by arrows). This process continued over time, as seen in images taken at 7 and 8 weeks (Figure 4d,e), as well as at 3.5 and 4.5 months post-surgery (Figure 4f,g). Corresponding X-rays at 4.5 months are shown in Figure 4h,i.

### 7.3. Reduction in Post-Operative Swelling Inflammation [107]

A 62-year-old man presented with degenerative changes in the first metatarsophalangeal joint (Hallux Rigidus) and metatarsalgia. His forefoot deformities included hallux valgus interphalangeus, subluxation of the lesser MTP joints, a hammer toe deformity of the second toe, and a bunionette deformity (Figure 5a). The surgical intervention involved a cheilectomy of the first metatarsal head, an Akin–Moberg osteotomy at the base of the proximal phalanx of the big toe, Weil osteotomies of the second and third metatarsals, a Chevron osteotomy of the fifth metatarsal, and PIPJ arthrodesis to correct the second hammer toe (Figure 5b).

Fixation was achieved using Kirschner wires (KWs) for the second toe and fifth metatarsal, screws for the second and third metatarsals, and an absorbable suture for the proximal phalanx osteotomy of the big toe. Following surgery, copper dressings were applied immediately, with the first dressing change performed after three weeks. At that time, the surgical wounds showed no signs of infection or inflammation (Figure 5c). Remarkably, despite undergoing four metatarsal osteotomies and a toe arthrodesis, there was minimal swelling, allowing the natural skin wrinkles to remain visible (Figure 5c–e). This outcome contrasts sharply with the significant swelling typically observed for several months following foot osteotomies.

### 7.4. Reduction in Scar Formation [107]

A 20-year-old healthy woman underwent bunion surgery, which involved a Chevron-type osteotomy of the first metatarsal with fixation using Kirschner wires (KWs) (Figure 6a). Two weeks post-surgery, both the surgical incision and the KWs were visible in clinical photos and X-rays (Figure 6a,b). The KWs were removed at four weeks, by which time the surgical incision had healed well (Figure 6c,d).

At seven weeks post-surgery, the osteotomy had fully healed, and the clinical outcome was satisfactory (Figure 6e,f). The surgical scar appeared very faint (Figure 6g). A comparison of the initial incision with its appearance at seven weeks revealed that approximately 80% of the scar was no longer visible, even under high-resolution and magnified photography (Figure 6h–j). This suggests that either direct epithelialization occurred or scar remodeling took place. The exceptional cosmetic results at seven weeks go beyond what is typically expected from a successful case, leading us to attribute this outcome to the positive effects of the copper dressings on wound healing.

## 8. Conclusions

Copper exhibits a broad spectrum of antimicrobial activity similar to silver. However, unlike silver, copper is an essential element for bodily functions and nutrition, and it is not cytotoxic. MedCu copper dressings incorporate microparticles of cuprous oxide, which act as a reservoir for copper ions. These ions are released gradually and consistently into the wound at parts-per-million (ppm) levels, maintaining an optimal concentration of copper ions. This controlled release stimulates the production of cytokines and growth factors, promoting angiogenesis and granulation tissue formation.

Additionally, copper ions act as cofactors for enzymes involved in wound healing, such as elastases and metalloproteinases. These enzymes facilitate the autolysis of necrotic tissue, contributing to copper dressings’ potent debridement effect. The presence of copper also enhances epithelial migration, accelerating wound closure and, in some cases, eliminating the need for skin graft surgeries.

All these actions occur simultaneously, with copper also providing antibacterial protection to the wound. As a result, copper supports all stages of wound healing and addresses the essential elements of this complex physiological process. This clinical observation, which is in line with basic science, led us to formulate the concept of a “Continuum of Care through all phases of wound healing”. This concept is important because it means that whenever the caregiver debates which treatment the wound needs now, with copper dressings, they cannot be wrong.

The growing number of independently conducted animal and laboratory studies, as described in Section 5, all demonstrating the positive effects of copper-containing biomaterials—primarily dressings—on wounds, reinforces our findings and supports the conclusion that copper-containing dressings will play a vital role in the expanding the “arsenal” of wound healing interventions available to practitioners.

## 9. Future Directions

In this article, we explored the use of copper dressings in managing chronic wounds. However, based on our clinical experience, we believe that copper-containing dressings can also be highly beneficial in treating acute wounds, such as surgical wounds, skin tears, abrasions, lacerations, and burns. In such cases, we have observed a significant reduction in wound complications, such as dehiscence and infections, as well as a noticeable decrease in scarring.

Given copper’s essential role in all physiological processes of wound healing, we have seen its beneficial effects across all stages of the healing process. The ability to use a single dressing effectively from the initial wound occurrence to full closure—particularly by general practitioners rather than specialized wound care experts—could greatly enhance the efficiency of wound management. This is especially important in developing countries, where wound specialists may be scarce and access to effective treatments is limited.

To further validate these observations, large-scale clinical trials involving a diverse range of wound types and indications should be conducted to clearly establish copper-containing dressings as a potential “game changer” in wound management.

## Figures and Tables

**Figure 1 biomedicines-13-00562-f001:**
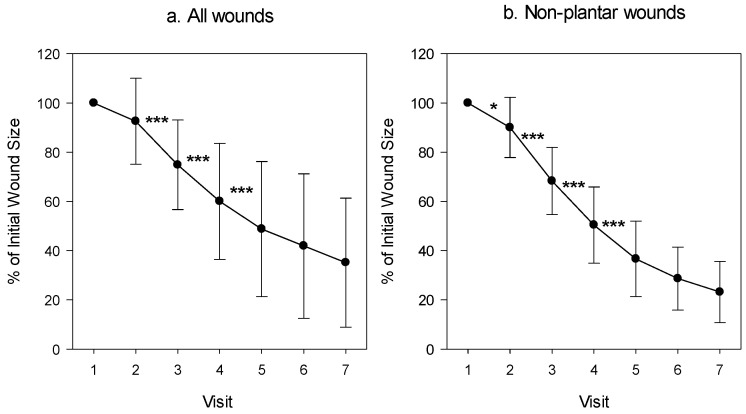
Percent of wound area during the trial as compared to the wound size at the commencement of the study. The mean ± standard deviation of the wound area of (**a**) all patients and (**b**) of those suffering from a non-plantar foot wound. A paired t-test was conducted between each visit and the subsequent one. * *p* < 0.05; *** *p* < 0.001 [110].

**Figure 2 biomedicines-13-00562-f002:**
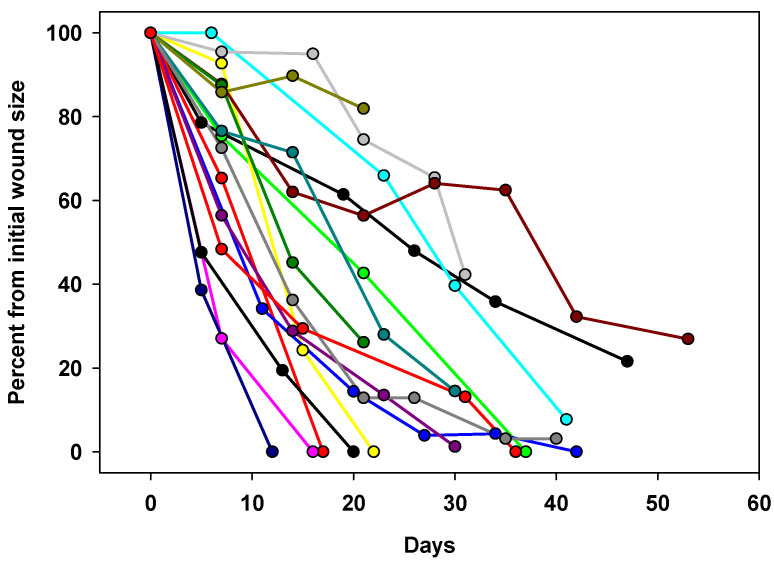
Wound healing progression during the use of the copper dressings. Each line represents the size of the wound per each patient after normalization to the start of the wound size at the commencement of the copper dressing use [113].

**Figure 3 biomedicines-13-00562-f003:**
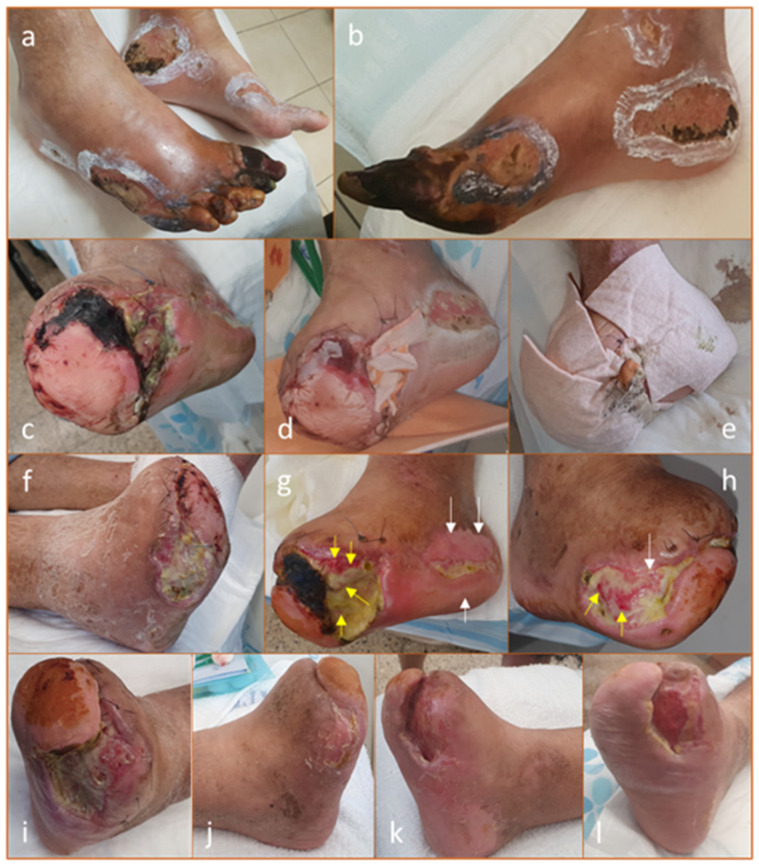
Resolution of Infection, Granulation Induction, and Epithelialization of Necrotic Wounds. (**a**). Ulcers colonized with *Pseudomonas aeruginosa* were present on both feet, predominantly on the right foot. (**b**). The ulcers were located on the medial aspect of the heel and the lateral side of the foot. (**c**). Two weeks post-transmetatarsal amputation, necrotic tissue was observed at the edges of the partially closed flap. (**d**). Copper dressings were first applied (Day 0), placed deep within the plantar-fascial region of the amputation wound. (**e**). Additionally, the dressings were used to cover both the medial and lateral ulcers. (**f**). After one-week, necrotic tissue had reduced, and granulation tissue formation had begun in all wounds. (**g**,**h**). After two months of copper dressing treatment, clear epithelialization (white arrows) was observed in the medial and lateral ulcers, while granulation tissue (yellow arrows) was evident in these ulcers and the central amputation wound, visible beneath a thin layer of residual necrotic tissue. Cultures from the necrotic tissue were negative for *Pseudomonas*, the initial resistant pathogen. (**i**). The granulation tissue appeared to facilitate necrotic tissue autolysis (self-debridement). (**j**,**k**). By five months of copper dressing treatment, the medial and lateral ulcers had completely closed. (**l**). The primary wound was partially closed, with the remaining area covered in healthy pink to red granulation tissue.

**Figure 4 biomedicines-13-00562-f004:**
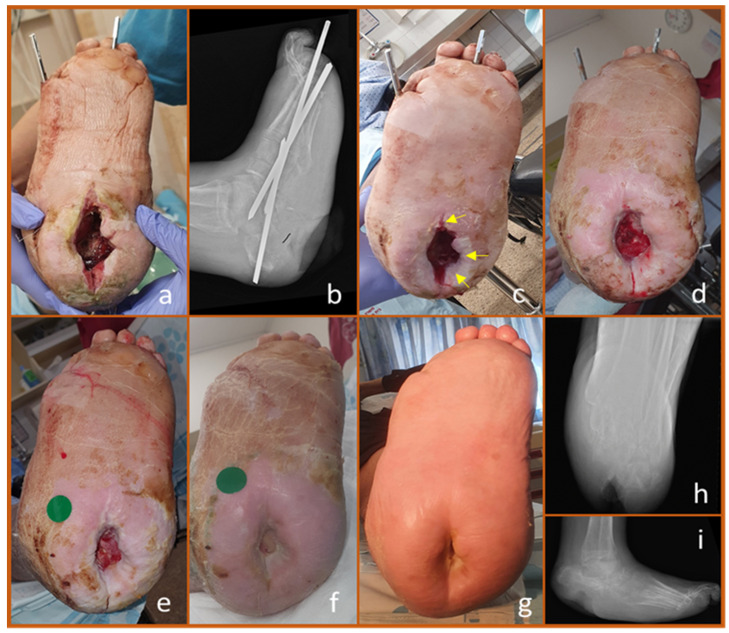
Epithelialization of a Plantar Deep Wound Associated with Rocker Deformity. The patient underwent resection of an infected calcaneal bone, followed by correction and stabilization of the resulting rocker deformity using Steinman pins. Copper dressings were applied during surgery. (**a**). One-week post-surgery, the deep calcaneal wound was visible without signs of infection. (**b**). X-ray showing the inserted Steinman pins, with the missing plantar calcaneal bone and the deep soft tissue void beneath it. (**c**). Three weeks after surgery, copper dressings treatment promoted the migration of epidermal tissue from the wound edges into its depth. (**d**,**e**). Continued epithelialization was observed over the granulation tissue at 7 weeks (**d**) and 8 weeks (**e**) of copper dressings treatment. (**f**). At 3.5 months, significant wound closure was evident. (**g**). By 4.5 months, the wound had achieved 95% closure. (**h**,**i**). Lateral and axial calcaneal X-rays at 4.5 months post-surgery revealed a large soft tissue void, now covered with nearly normal-appearing skin. Throughout the entire period, the foot was treated exclusively with copper dressings.

**Figure 5 biomedicines-13-00562-f005:**
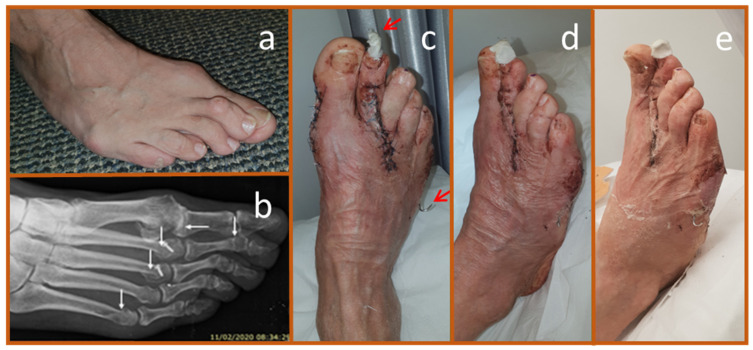
Reduction of Swelling Following Forefoot Surgeries and Osteotomies. (**a**). Preoperative forefoot deformities, including hallux valgus, hammer toe of the second digit, subluxation of the 2nd and 3rd metatarsophalangeal joints, and a Bunionette deformity. (**b**). X-ray taken two months post-surgery showing osteotomy of the proximal phalanx base of the big toe, Weil osteotomies of the 2nd and 3rd metatarsals, Chevron osteotomy of the 5th metatarsal, and PIPJ arthrodesis of the 2nd hammer toe (arrows). (**c**). Clinical image at the first dressing change, three weeks after surgery (copper dressings were applied intraoperatively), showing well-healing surgical wounds without signs of inflammation. The absence of swelling and the presence of skin wrinkles are notable. The tips of the K-wires stabilizing the 2nd toe and 5th metatarsal are visible (marked with red arrows); the second toe K-wire is wrapped in plaster to prevent accidental pullout. (**d**). Oblique view of the same foot at the same visit, after suture removal, clearly demonstrating reduced swelling. (**e**). Clinical images at five weeks post-surgery showing maintained skin wrinkles and continued absence of swelling.

**Figure 6 biomedicines-13-00562-f006:**
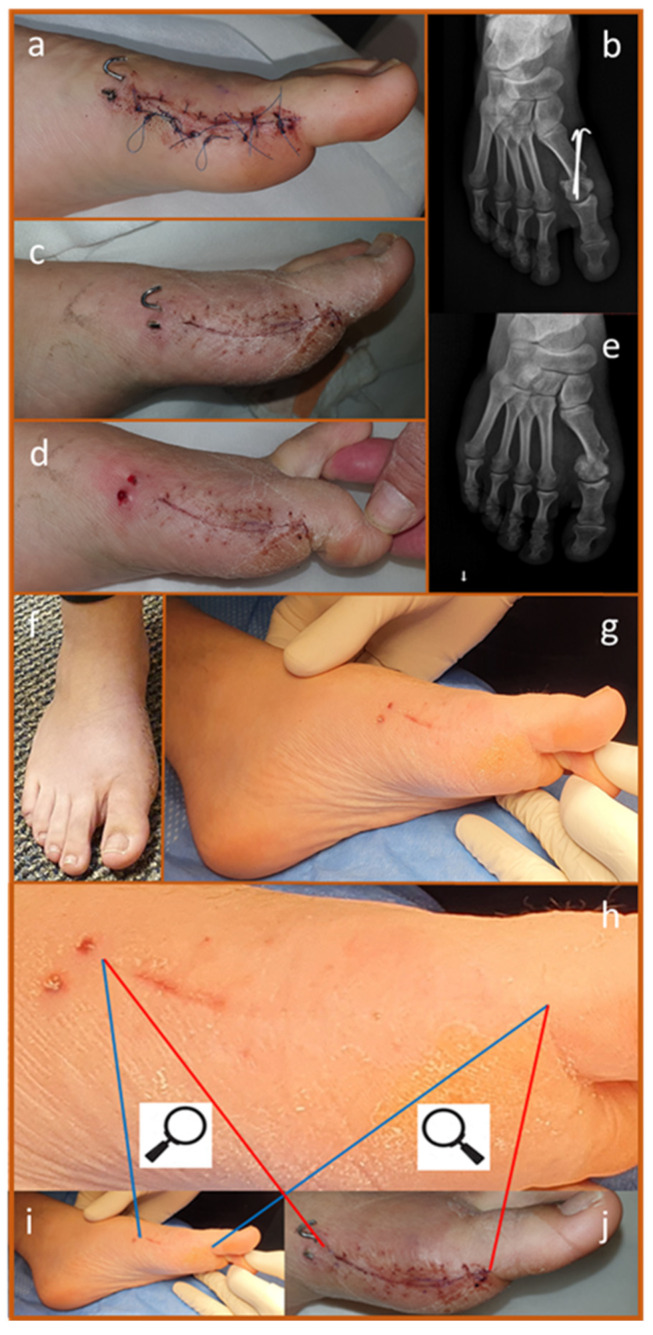
(**a**,**b**). Clinical and radiographic images taken two weeks post-surgery, showing the surgical incision and the K-wires in place. (**c**,**d**). By four weeks, the K-wires were removed, and the surgical incision showed good healing. (**e**,**f**). At seven weeks post-surgery, the osteotomy had fully healed, with a satisfactory clinical outcome. (**g**). The surgical scar appeared very delicate. (**h**–**j**). Comparison between the original incision and its appearance at seven weeks demonstrated that approximately 80% of the scar was no longer visible, even under high-resolution magnification. This suggests either direct epithelialization or significant scar remodeling.

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
