# Peer review of "The Journey of Copper-Impregnated Dressings in Wound Healing: From a Medical Hypothesis to Clinical Practice"

_biomedicines, 2025, doi:10.3390/biomedicines13030562_

Round 1

Reviewer 1 Report

Comments and Suggestions for Authors

Comments for the authors:

The manuscript under revision entitled “The Journey of Copper-Impregnated Dressings in Wound Healing: From a Medical Hypothesis to Clinical Practice” provides a comprehensive and detailed review of the role of copper dressings in the management of chronic wounds. The authors provide a comprehensive overview of the scientific background, laboratory findings and clinical applications of copper dressings and highlight the advantages in the area over traditional wound care treatments.

The article is well referenced and provides valuable insights into an emerging biomedical technology. However, several aspects require further clarification and improvement to increase its impact and rigorist.

Major Comments:

1-     In the abstract, the authors state that chronic wounds are associated with diabetes, but there are many diseases associated with chronic wounds that the authors describe in the introduction. Correct this sentence in the abstract.

2-     The authors declared an affiliation with MedCu Technologies, the manufacturer of the copper dressings. While this does not invalidate the study, a clearer statement of how potential bias was managed is needed. A discussion of independent studies evaluating copper dressings would strengthen the credibility of the manuscript.

3-     Details of the methodology used in the preclinical and clinical evaluations should be more explicit. A brief summary of the study design parameters (e.g. wound size, treatment duration, outcome measures) would also be useful.

4-     Details of the trial and, in particular, the trade name of the dressings should be provided. Patient consent is also required to address any ethical concerns.

5-     The paper mainly compares copper dressings with silver dressings. However, there is no analysis of other advanced wound care technologies such as growth factor-based therapies, bioengineered skin substitutes and stem cell therapies. The quality of the manuscript would also be greatly improved by a discussion of the limitations of copper dressings.

6-     Including a discussion of existing large-scale clinical trials (if available) or acknowledging the need for such trials would improve the scientific robustness of the manuscript.

Minor Comments:

1-     Check grammar. Improve some English expressions

2-     Introduction could be shortened.

Final Decision:

Although the article is of interest in the field of materials and shows clear evidence of the benefits of copper-based dressings, some important aspects mentioned in the “Major Reviews” section should be considered. I believe that the article can be published in Biomedicines after these Major Revisions have been addressed by the authors.

Comments on the Quality of English Language

See comments above.

Author Response

The manuscript under revision entitled “The Journey of Copper-Impregnated Dressings in Wound Healing: From a Medical Hypothesis to Clinical Practice” provides a comprehensive and detailed review of the role of copper dressings in the management of chronic wounds. The authors provide a comprehensive overview of the scientific background, laboratory findings and clinical applications of copper dressings and highlight the advantages in the area over traditional wound care treatments.

The article is well referenced and provides valuable insights into an emerging biomedical technology.

              Response: Thank you.

However, several aspects require further clarification and improvement to increase its impact and rigorist.

Major Comments:

1-     In the abstract, the authors state that chronic wounds are associated with diabetes, but there are many diseases associated with chronic wounds that the authors describe in the introduction. Correct this sentence in the abstract.

Response: The sentence was corrected as suggested. It now reads “Chronic wounds pose a substantial global healthcare burden, exacerbated by aging populations and the increasing prevalence of conditions such as diabetes, peripheral vascular disease, and venous insufficiency.”.

2-     The authors declared an affiliation with MedCu Technologies, the manufacturer of the copper dressings. While this does not invalidate the study, a clearer statement of how potential bias was managed is needed. A discussion of independent studies evaluating copper dressings would strengthen the credibility of the manuscript.

Response: Section 5, called “Further laboratory and animal studies supporting the key role that copper plays in wound healing” refers to 42 articles published not by the authors of this article. All these articles strongly support the very beneficial role that wound dressings containing copper have on wound healing. All these studies independently support the credibility of our findings. We have added now in the Conclusion Section the following paragraph: “The growing number of independently conducted animal and laboratory studies, as described in Section 5, all demonstrating the positive effects of copper-containing biomaterials—primarily dressings—on wounds, reinforces our findings and supports the conclusion that copper-containing dressings will play a vital role in the expanding "arsenal" of wound healing interventions available to practitioners.”.

3-     Details of the methodology used in the preclinical and clinical evaluations should be more explicit. A brief summary of the study design parameters (e.g. wound size, treatment duration, outcome measures) would also be useful.

Response: As suggested, we have added a brief summary and study details, including relevant graphs, regarding three clinical studies conducted in non-infected wounds. These studies clearly support the notion that the wound-healing benefits of the copper dressings extend beyond their antimicrobial properties. This is presented in the revised Section 6.

4-     Details of the trial and, in particular, the trade name of the dressings should be provided. Patient consent is also required to address any ethical concerns.

Response: As required by the Editor, since the article is a review article and not a description of a particular trial, the unpublished data (Section 7 – Case Samples) was removed, and replaced with 4 already published cases and thus the requirement of patient consent is not relevant for the revised article. The trade name of the dressings is now given in the last paragraph of Section 5.

5-     The paper mainly compares copper dressings with silver dressings. However, there is no analysis of other advanced wound care technologies such as growth factor-based therapies, bioengineered skin substitutes and stem cell therapies. The quality of the manuscript would also be greatly improved by a discussion of the limitations of copper dressings.

Response: The reason we relate to silver dressings, which comprise more than 70% of the antimicrobial wound dressing global market, is because both silver and copper have similar wide spectrum antimicrobial properties. But, as opposed to silver, copper is an essential trace element involved in many of the wound healing physiological processes, and the comparison made to silver dressings is to emphasize that the benefits of the copper dressings extend beyond their antimicrobial properties. There are many advanced wound care technologies, but making a comparison with such technologies is not the objective of this manuscript. Such an analysis deserves a separate comprehensive review, which we intend doing in the future.  

6-     Including a discussion of existing large-scale clinical trials (if available) or acknowledging the need for such trials would improve the scientific robustness of the manuscript.

Response: Unfortunately, there are no existing large trials with the relatively new copper dressings. Accordingly, we have added in the last part of the article following sentence: ”To further validate these observations, large-scale clinical trials involving a diverse range of wound types and indications should be conducted to clearly establish copper-containing dressings as a potential "game changer" in wound management.”.  

Minor Comments:

  • Check grammar. Improve some English expressions

Response: Done

  • Introduction could be shortened.

Response: Done

Reviewer 2 Report

Comments and Suggestions for Authors

The article explores the role of copper-impregnated dressings in enhancing wound healing, emphasizing their ability to stimulate angiogenesis, granulation tissue formation, and antimicrobial protection. It highlights the controlled release of copper ions to address chronic wounds effectively and demonstrates superior outcomes compared to conventional dressings, including silver-based ones. Clinical applications showcase their potential as a cost-effective, innovative solution for various types of wounds. It is well written and organized. I suggest its publication after minor changes

1.    Consider adding a graphical summary or flowchart to visualize the mechanisms of copper in wound healing and its clinical applications.

2.    Explore the possibility of including more quantitative data from case studies or clinical trials to support the claims more robustly.

3.    Discuss potential side effects or contraindications of copper dressings, even if minor, to provide a balanced view.

4.    Where are the figures taken from? If they are taken from published articles permission is needed.

5.    It is worth a section like Future Directions.

6.    Please include some quantitative data and graphs from articles.

Author Response

The article explores the role of copper-impregnated dressings in enhancing wound healing, emphasizing their ability to stimulate angiogenesis, granulation tissue formation, and antimicrobial protection. It highlights the controlled release of copper ions to address chronic wounds effectively and demonstrates superior outcomes compared to conventional dressings, including silver-based ones. Clinical applications showcase their potential as a cost-effective, innovative solution for various types of wounds. It is well written and organized. I suggest its publication after minor changes.

              Response: Thank you.

  1. Consider adding a graphical summary or flowchart to visualize the mechanisms of copper in wound healing and its clinical applications.

              Response: A graphical summary was added at the end of the article.

  1. Explore the possibility of including more quantitative data from case studies or clinical trials to support the claims more robustly.

              Response: As suggested, more quantitative data has been added to describe the results obtained in the clinical trials in Section 6, including relevant graphs.

  1. Discuss potential side effects or contraindications of copper dressings, even if minor, to provide a balanced view.

              Response: We estimate that the Copper Dressings have been used in more than 100,000 wounds without any adverse events being reported, indicating their very high safety. We have now added the following sentence in Section 6: “The use of copper containing dressings should be avoided in cases of known allergy to copper or in oncological active wounds.”.

  1. Where are the figures taken from? If they are taken from published articles permission is needed.

              Response: The figures were original unpublished figures. The Editor asked to replace them with already published figures, as this is a review article and original figures are not allowed. We have thus replaced the figures with already published figures after obtaining permission and quoting the relevant references.

  1. It is worth a section like Future Directions.

              Response: A new Future Directions section was added at the end of the manuscript.

  1. Please include some quantitative data and graphs from articles.

              Response: As suggested, quantitative and graphs from the quoted articles have been added to the revised Section 6.

Reviewer 3 Report

Comments and Suggestions for Authors

For Authors:

Suggestions and Comments for the Authors:

In my opinion, the review is well-designed and provides very good information. Below, I suggest some comments to improve the review:

  1. In Section 1, please add more elements such as silver, which you mentioned for wound healing. For example, discuss whether cobalt is good for wound healing. A small discussion will improve this section.
  2. Please add some data about the role of doping copper in bioactive glass in Section 5 for improving wound healing. Using both copper and bioactive glass is recently suggested as a powerful technique for improving wound healing.
  3. In Section 5, please add more data about the size of copper particles, as about one-third of your references discuss the role of nanomaterials.
  4. On page 12, please enlarge the note in Figure E.
  5. I think the conclusion needs some improvement. Please enhance it.

Good luck!

Author Response

In my opinion, the review is well-designed and provides very good information. Below, I suggest some comments to improve the review:

  1. In Section 1, please add more elements such as silver, which you mentioned for wound healing. For example, discuss whether cobalt is good for wound healing. A small discussion will improve this section.

Response: Section 1 describes chronic wounds and physiological healing processes. As suggested, we have added a small discussion for cobalt in Section 2, as we think it is a more appropriate section.

  1. Please add some data about the role of doping copper in bioactive glass in Section 5 for improving wound healing. Using both copper and bioactive glass is recently suggested as a powerful technique for improving wound healing.

Response: Done as suggested, also in relation to cobalt.

  1. In Section 5, please add more data about the size of copper particles, as about one-third of your references discuss the role of nanomaterials.

Response: Done as suggested.

  1. On page 12, please enlarge the note in Figure E.

Response: The figures were original unpublished figures. The Editor asked to replace them with already published figures, as this is a review article and original figures are not allowed. We have thus replaced the figures with already published figures after obtaining permission and quoting the relevant references.

  1. I think the conclusion needs some improvement. Please enhance it.

Response: Done as suggested.

Round 2

Reviewer 1 Report

Comments and Suggestions for Authors

Authors respond satisfactorily to all questions asked by the reviewer. The additions and corrections made significantly improve the quality and the accuracy of the manuscript.

I recommend that the manuscript be accepted for publication in Biomedicines in its present form.